

# Parvovirus B19 DNA and antibodies in Chinese plasma donors, plasma pools and plasma derivatives

Pan Sun[1,*], Peng Jiang[1,*], Qing Liu[1], Rong Zhang[1], Zongkui Wang[1], Haijun Cao[1], Xiangzhong Ye[2], Shangzhi Ji[2], Jinle Han[2], Kuilin Lu[3], Xuexin He[3], Jiajin Fan[4], Dawei Cao[4], Yu Zhang[5], Yongsheng Yin[5], Yunhua Chen[6], Xuemei Zhao[6], Shengliang Ye[1], Na Su[1], Xi Du[1], Li Ma[1] and Changqing Li[1]

[1] Institute of Blood Transfusion, Chinese Academy of Medical Sciences, Chengdu, China
[2] Beijing Wantai Biological Pharmacy, Beijing, China
[3] Chengdu Rongsheng Pharmaceutical Co., Ltd., Chengdu, China
[4] Shandong Taibang Biological Products Co., Ltd., Taian, China
[5] Hualan Biological Products Co., Ltd., Xinxiang, China
[6] Guizhou Taibang Biological Products Co., Ltd, Guiyang, China
[*] These authors contributed equally to this work.

Corresponding authors
Li Ma, mary@ibt.pumc.edu.cn
Changqing Li,
lichangqing268@163.com

## ABSTRACT

**Background**. Human parvovirus B19 (B19V) is a common contaminant found in plasma pools and plasma derivatives. Previous studies were mainly focused on limited aspects, further assessment of prevalence of B19V DNA and antibodies in plasma donors, the contamination of B19V in pooled plasma and plasma derivatives should be performed in China.

**Study Design and Methods**. Individual plasma donors' samples from four provinces and pooled plasma from four Chinese blood product manufacturers were collected and screened using B19V DNA diagnostic kits between October 2018 and May 2020. The positive samples were investigated for the seroprevalence of B19V antibodies and subjected to sequence analysis and alignment for phylogenetic studies. Moreover, 11 plasma donors who were B19V DNA-positive at their first testing were also followed during the later donation period. Additionally, 400 plasma pools and 20 batches of plasma derivatives produced by pooled plasma with a viral load of B19V DNA exceeding $10^4$ IU/mL were also collected and tested for B19V DNA and antibodies.

**Objectives**. To comprehensively and systematically determine the frequency and viral load of B19V DNA in plasma donors, pooled plasma, and plasma derivatives from four Chinese blood product manufacturers.

**Results**. A total of 17,187 plasma donors were analyzed and 44 (0.26%) specimens were found positive for B19V DNA. The quantitative DNA levels ranged from $1.01 \times 10^1$ to $5.09 \times 10^{12}$ IU/mL. Forty-four DNA-positive specimens were also investigated for the seroprevalence of B19V antibodies, 75.0% and 2.3% of which were seropositive for B19V IgG and IgM antibodies, respectively. The phylogenic analyses showed that the prevalent genotypes in the four provinces' plasma donors belonged to B19V Genotype 1. Eleven individual plasma donors who were B19V DNA-positive at the first donation were then followed for a period, and in general, the DNA levels of B19V gradually decreased. Moreover, 64.8% (259/400) of the pooled plasma was contaminated by B19V, with concentrations of $1.05 \times 10^0$–$3.36 \times 10^9$ IU/mL. Approximately 72.6%

of the DNA-positive plasma pools were only moderately contaminated ($<10^4$ IU/mL), while 27.4% contained $>10^4$ IU/mL. Twenty batches of plasma derivatives produced by pooled plasma with a viral load of B19V DNA exceeding $10^4$ IU/mL were also tested. B19V was detected in 5/5 PCC samples and 5/5 factor VIII samples but was not found in the intravenous immune globulin and albumin samples.

**Conclusion**. The contamination of B19V in pooled plasma and plasma-derived clotting factor concentrates is serious. Whether B19V nucleic acid testing (NAT) screening of plasma and plasma derivatives is launched in China, blood product manufacturers should spontaneously perform B19V NAT screening in plasma donors and mini-pool plasma. These measures can ensure that samples with high titer B19V DNA are discarded in order to prevent and control this transfusion transmitted virus.

# INTRODUCTION

Human parvovirus B19 (B19V) is a non-enveloped virus with a liner single-stranded DNA genome. It belongs to Erythroparvovirus of the *Parvoviridae* family. B19V infection causes a variety of illnesses, including Fifth disease in children, aplastic crisis in patients with hemolytic disorders, fatal hydrops in pregnant women, arthropathy, cardiomyopathy, and inflammation of other various tissues (*Qiu, Söderlund-Venermo & Young, 2017*). Moreover, the epidemiology of B19V shows a wide geographic distribution and seasonal variations, and the peak season of infection is spring and winter. The transmission of B19V is primarily through the upper respiratory route, organ transplantation, and blood transfusion (*Ganaie & Qiu, 2018*; *Parsyan & Candotti, 2007*). Since the prevalence of B19V in blood donors is about 1% (*Abdelrahman et al., 2021*), there is a high probability that plasma pools can also be contaminated. When DNA in the blood donor population reaches 1%, their plasma could easily contaminate the entire pooled plasma. Once a single plasma donor with extremely high DNA concentrations exceeding $10^{14}$ IU/mL, the B19 viral DNA in the pooled plasma could reach $10^{10}$ IU/ml. Because of its special structure and small size, B19V is highly resistant to all commonly-used inactivation methods. Our preliminary research also indicated that plasma-derived clotting factor concentrates such as fibrinogen and PCC were found to be highly contaminated with B19V DNA (53.7%–85.7%), and IVIG and albumin were moderately contaminated (0–38.9%) (*Zhang et al., 2012*). Therefore, the safety of plasma products needs to be further evaluated.

In China, there has been no specific documentation or technical guidelines for monitoring B19V. To better ensure the safety of plasma derivatives, many governments request plasma manufacturers perform B19V screening in manufacturing plasma pools. For example, the US Food and Drug administration guidelines, US Pharmacopoeia, Plasma Protein Therapeutics Association, and European regulatory requirements recommend that the viral load of B19V in the manufacturing of pooled plasma should not exceed
$10^4$IU/mL (*The European Pharmacopoeia, 2020a*; *The European Pharmacopoeia, 2020b*; *Jin et al., 2010*). Although the B19V-DNA prevalence among Chinese plasma donors is relatively low, asymptomatic plasma donors with high levels of B19V (up to $10^{12}$ IU/mL) may present with a greater risk in plasma derivatives (*Jia et al., 2019*; *Li et al., 2020*; *Marano et al., 2015*; *Schmidt et al., 2001*; *Siegl & Cassinotti, 1998*; *Zhang et al., 2012*). This study's major objective was to comprehensively and systematically determine the frequency and levels of B19V DNA in plasma donors, pooled plasma, and plasma derivatives from four Chinese blood products manufacturers. Additionally, reports on changes in B19V DNA in single–plasma donors are rare, so we also followed-up and evaluated changes in B19V DNA level in those B19V DNA-positive donors who were first confirmed after their subsequent donation. We continued to track and investigate contamination in the plasma derivatives produced by plasma pools with viral load of B19V DNA exceeding $10^4$ IU/mL. Moreover, B19V-specific immunoglobulin (IgG/IgM) antibodies in all of B19V DNA-positive samples were also determined. We expect that this study will provide a series of robust evidence on promoting the quality standards for plasma derivatives in China.

## MATERIALS & METHODS

### Sample collection

From October 2018 through May 2020, 17,380 individual plasma samples from four provinces (Shandong, Guangxi, Sichuan, and Guizhou) and 400 plasma pools (approximately 4,000 single-plasma mixed a pool) were collected from four Chinese blood product manufacturers. Furthermore, we collected samples from 11 B19V DNA-positive donors who were frequent plasma donors after their first DNA-positive donation for a period ranging from 2 months to 6 months. Additionally, we collected five batches of albumin, five batches of intravenous immunoglobulin, five batches of prothrombin-complex concentrate (PCC), and five batches of Factor VIII produced by plasma pools with a viral load of B19V DNA exceeding $10^4$IU/mL from company B. The study was approved by the Research Ethics Committee of the Institute of Blood Transfusion (No. 202029). The study was based on the plasma samples stored in place collected before and the researchers had no direct contact with the donors. The Ethics Committee waived the need for written consent.

### B19V DNA quantitation

Individual plasma samples were tested by pools of 48 individuals. The samples in the B19V positive pools and the pooled plasma were tested separately with 960 μL plasma. A virus DNA/RNA kit (Beijing Wantai Biological Pharmacy Enterprise Co., Ltd, Beijing, China) was used for nucleic acid extraction according to the manufacturer's instructions. The DNA extracts were stored at −80 °C prior to PCR analysis. B19V DNA samples were screened using human parvovirus B19V DNA diagnostic kits (PCR-fluorescence probing) (Beijing Wantai Biological Pharmacy Enterprise Co., Ltd, Beijing, China). This kit can detect all three B19V genotypes with a sensitivity of 20 IU per mL. The Q-PCR assays were performed on a Bio-Rad CFX96 real-time PCR platform (Bio-Rad Laboratories, Hercules, CA, USA).

## Phylogenetic analysis

The phylogenetic information was analyzed using the sequence from NS1-VP1-U region. The sequence was amplified according to *Servant et al. (2002)*. The PCR products were purified using NucleoSpin Extract II kit (Macherey-Nagel GmbH & Co. KG, Duren, Germany) according to the instructions. The second cycle sequencing reactions were performed using the purified products. The sequences were then read with ABI 3730 (Applied Biosystems, Waltham, MA, USA). Sequences were determined for both directions. The sequences were aligned using ClustalX 1.83. The neighborhood joint (NJ) and maximum parsimony (MP) analyses were used to detect the phylogenetic position of the samples in this study with the reference sequences using software MEGA 7.0.

## B19V serologic assays

B19V-specific antibodies in B19V DNA-positive specimens were investigated using commercial assay kits (Virion-Serion, Würzburg, Germany) according to the manufacturer's instructions.

## Descriptive statistics

Data collection was performed using Microsoft Excel 2011, and descriptive statistical analysis was performed using SPSS 16.0 statistics software (SPSS, Inc., Chicago, IL, USA).

# RESULTS

## Prevalence of B19V DNA and antibodies in plasma donors

In this study, 17,187 individual samples from plasma donors living in four Chinese provinces were collected and tested. Of 17,187 plasma donors, 44 (0.26%) specimens were found positive for B19V DNA. The quantitative DNA levels ranged from $1.01 \times 10^1$ IU/mL to $5.09 \times 10^{12}$ IU/mL. As shown in Table 1, one of 2,000 (0.05%) individual plasma samples from company A was positive for B19V DNA. Eighteen of 3,898 (0.46%) individual plasma samples from company B were positive for B19V DNA, and two positive (0.05%) samples contained $>10^4$ IU/mL. Company C and Company D had 0.33% (13/3985) and 0.16% (12/7304) B19V DNA-positive samples, respectively. Moreover, the distribution of ABO blood types in B19-postive donors was different (blood type A was 18.2%, blood type B was 22.7%, blood type AB was 6.8%, and blood type O was 52.3%). The anti-B19V IgG and IgM titers were also tested in B19V DNA-positive samples. Twenty-eight out of 44 (63.6%) samples were positive for IgG, with titers in the range of 7.03-3,800 IU/mL. Only one sample of the 44 (2.3%) contained IgM, with a titer of 30.91 IU/mL. Eleven individual plasma donors who were B19V DNA positive at their first donation in this study were followed during their later donations (minimum period was two months, and maximum period was six months). The B19V DNA, IgG, and IgM were monitored during the following donations until the COVID-19 pandemic outbreak in 2020. The follow-up tests are shown in Table 2. Eight of 11 (72.7%) donors were positive for B19V DNA no more than two times during the following period. Two donors were positive for B19V DNA across the following periods, while the other donors were positive for B19V DNA six out of nine donations during the 6 following months. The donor with the highest B19V

**Table 1  B19V DNA prevalence in plasma donors from four Chinese provinces between 2018 and 2021.**

| Blood products manufactures | Areas of plasma samples | Samples detection (n) | B19 DNA positive (n) | Prevalence of B19V DNA (%) | 95% confidence interval [CI] |
|---|---|---|---|---|---|
| A | Sichuan | 2,000 | 1 | 0.05 | $0.00 \sim 0.15$ |
| B | Guangxi | 3,898 | 18 | 0.46 | $0.24 \sim 0.66$ |
| C | Shandong | 3,985 | 13 | 0.33 | $0.15 \sim 0.50$ |
| D | Guizhou | 7,304 | 12 | 0.16 | $0.07 \sim 0.26$ |
| Total | / | 17,187 | 44 | 0.26 | $0.18 \sim 0.33$ |

DNA ($5.09 \times 10^{12}$ IU/mL) was negative for IgG/IgM at the first donation after index time, IgG and IgM reached their peak titers at the second donation, IgM became negative a half month later, and IgG remained positive during the following periods.

### B19V DNA and antibodies in plasma pools for fractionation

A total of 400 plasma pools were tested for B19V DNA, IgG, and IgM. Of these, 259 pools (64.8%) contained B19V DNA, and 76 out of the 259 (29.3%) contained B19V DNA at a level higher than $10^4$ IU/mL. The prevalence of B19V DNA in plasma pools differed across companies. Company A had a prevalence of 27% while the other three companies had much higher prevalence of B19V DNA (Company B 90%, Company C 81%, and Company D 61%). Meanwhile the number of plasma pools with a viral load higher than $10^4$ IU/mL differed across the four companies. All the B19V DNA-positive plasma pools were positive for B19V IgG, with titers in the range of 5.755–27.73 IU/mL. No sample was positive for B19V IgM. The results are shown in Table 3.

### B19V DNA and antibodies in plasma derivatives

Plasma derivatives made from Company B plasma pools with a viral load of B19V DNA exceeding $10^4$ IU/mL were also collected and tested. Table 4 shows B19V DNA and antibodies in the plasma derivatives. B19V DNA was not detected in albumin and IVIG. However, all of the PCC and factors contained B19V DNA, and 80% (four of five batches) of PCC were highly contaminated (higher than $10^4$ IU/mL), and the B19V DNA concentration was as high as $3.59 \times 10^7$ IU/mL. Of all the plasma derivatives, only two batches of IVIG were positive for B19V IgG.

### Phylogenetic relationships among different B19V isolates

Seventeen 1,200-bp sample sequences were obtained within the NS1-VP1-U region. They were aligned with 35 reference sequences to reconstruct the phylogenetic tree under the NJ method. The tree is shown in Fig. 1 with the NJ bootstrap values depicted on the branches. All of the samples studied in this article formed a monophyletic group, and they fell in Genotype 1A.

## DISCUSSION

Because of its special raw resource, the safety of plasma derivatives is always an important and concerning issue (*Di Minno et al., 2016*). As a potential contaminant of blood

**Table 2  Varieties of viral load and antibody level in B19V DNA-positive donors during their different plasma donations.**

| Province | Donor | Sex | Age | Blood group | Plasma donation date | Viral load | IgM | IgG |
|---|---|---|---|---|---|---|---|---|
| Guangxi | 1 | Male | 18 | O | 20191018 | 5.09E+12 | − | − |
| | | | | | 20191104 | 1.86 E+05 | + | + |
| | | | | | 20191118 | 1.09E+04 | + | + |
| | | | | | 20191202 | 5.15 E+03 | − | + |
| | | | | | 20191216 | 4.96 E+03 | − | + |
| | | | | | 20191231 | 3.34 E+03 | − | + |
| | | | | | 20200115 | 3.04 E+03 | − | + |
| Guangxi | 2 | Female | 50 | O | 20191018 | 2.05 E+03 | − | − |
| | | | | | 20191101 | N/A | − | − |
| | | | | | 20191117 | N/A | − | − |
| | | | | | 20191201 | N/A | − | − |
| | | | | | 20191216 | N/A | − | − |
| Guangxi | | | | | 20191230 | N/A | − | − |
| | | | | | 20200113 | N/A | − | − |
| | | | | | 20200324 | N/A | − | − |
| | | | | | 20200407 | N/A | − | − |
| | | | | | 20200425 | N/A | − | − |
| Guangxi | 3 | Male | 54 | O | 20191022 | 1.28E+03 | − | + |
| | | | | | 20200328 | N/A | − | + |
| Guangxi | | | | | 20200411 | N/A | − | + |
| | | | | | 20200428 | N/A | − | + |
| Guangxi | 4 | Male | 23 | O | 20191017 | 1.06 E+03 | − | + |
| | | | | | 20191201 | 1.58 E+03 | − | + |
| | | | | | 20191215 | 1.39 E+03 | − | + |
| | | | | | 20200115 | 6.73 E+02 | − | + |
| | | | | | 20200323 | 4.61 E+02 | − | + |
| Guangxi | 5 | Female | 46 | B | 20191018 | 1.68 E+02 | − | + |
| | | | | | 20191101 | 5.50 E+01 | − | + |
| | | | | | 20191115 | 1.91 E+01 | − | + |
| | | | | | 20191129 | N/A | − | + |
| | | | | | 20191215 | 5.03 E+01 | − | + |
| Guangxi | | | | | 20191229 | N/A | − | + |
| | | | | | 20200112 | 1.25 E+01 | − | + |
| | | | | | 20200318 | 2.17 E+01 | − | + |
| Guangxi | | | | | 20200414 | N/A | − | + |
| | | | | | 20191018 | 7.12 E+01 | − | − |
| | | | | | 20191101 | N/A | − | − |
| | | | | | 20191115 | N/A | − | − |
| | | | | | 20191129 | 1.40 E+01 | − | − |

**Table 2** (*continued*)

| Province | Donor | Sex | Age | Blood group | Plasma donation date | Viral load | IgM | IgG |
|---|---|---|---|---|---|---|---|---|
| Guangxi | 6 | Female | 55 | A | 20191213 | N/A | − | − |
| | | | | | 20191227 | N/A | − | − |
| | | | | | 20200111 | N/A | − | − |
| | | | | | 20200330 | N/A | − | − |
| | | | | | 20200413 | N/A | − | − |
| | | | | | 20200427 | N/A | − | − |
| Guangxi | 7 | Male | 36 | B | 20191021 | 5.04 E+01 | − | − |
| | | | | | 20191115 | N/A | − | − |
| | | | | | 20191119 | N/A | − | − |
| | | | | | 20191204 | N/A | − | − |
| | | | | | 20191219 | N/A | − | − |
| | | | | | 20200102 | N/A | − | − |
| Guangxi | | | | | 20200116 | N/A | − | − |
| | | | | | 20200320 | N/A | − | − |
| | | | | | 20200404 | N/A | − | − |
| | | | | | 20200418 | N/A | − | − |
| | | | | | 20200502 | N/A | − | − |
| Guangxi | 8 | Female | 38 | AB | 20191018 | 4.12 E+01 | − | − |
| | | | | | 20191103 | N/A | − | − |
| | | | | | 20191120 | N/A | − | − |
| | | | | | 20191204 | N/A | − | − |
| | | | | | 20191218 | N/A | − | − |
| | | | | | 20200103 | N/A | − | − |
| Guangxi | | | | | 20200117 | N/A | − | − |
| | | | | | 20200321 | N/A | − | − |
| | | | | | 20200404 | N/A | − | − |
| | | | | | 20200419 | N/A | − | − |
| Guangxi | 9 | Male | 55 | B | 20191018 | 1.59 E+01 | − | + |
| | | | | | 20191109 | 1.43 E+01 | − | + |
| | | | | | 20191205 | N/A | − | + |
| | 10 | Female | 42 | O | 20191223 | N/A | − | + |
| | | | | | 20191020 | 1.46 E+01 | − | − |
| | | | | | 20191104 | N/A | − | − |
| | | | | | 20191118 | N/A | − | − |
| | | | | | 20191202 | N/A | − | − |
| Guangxi | | | | | 20191216 | N/A | − | − |
| | | | | | 20191230 | N/A | − | − |
| | | | | | 20200113 | N/A | − | − |
| | | | | | 20200318 | N/A | − | − |
| | | | | | 20200401 | N/A | − | − |
| | | | | | 20200415 | N/A | − | − |
| Guangxi | | | | | 20200429 | N/A | − | − |

**Table 2** (*continued*)

| Province | Donor | Sex | Age | Blood group | Plasma donation date | Viral load | IgM | IgG |
|---|---|---|---|---|---|---|---|---|
| Guangxi | 11 | Female | 44 | O | 20191018 | 1.01 E+01 | − | + |
| | | | | | 20191122 | N/A | − | + |
| | | | | | 20191207 | N/A | − | + |
| | | | | | 20191222 | N/A | − | + |
| | | | | | 20200107 | N/A | − | + |

transfusion disease, B19V can cause serious complications in some high-risk patients, such as those with hemophilia, potential hematological malignancy, and immunodeficiency. Due to B19V's pathogenicity and risk of transmission through plasma derivatives, great attention has been focused on it. Japan, Germany, and the Netherlands screen for B19V DNA or B19V specific antibodies in blood donors (*Groeneveld & van der Noordaa, 2003*; *Sakata et al., 2013*; *Schmidt et al., 2007*). In regards to plasma, the US Food and Drug Administration (FDA) and European Pharmacopoeia have proposed a limit of $10^4$IU/mL for B19V DNA levels in plasma pools when manufacturing all kinds of plasma derivatives (*The European Pharmacopoeia, 2020a*; *The European Pharmacopoeia, 2020b*; *Jin et al., 2010*). Although there are no regulations for monitoring B19V DNA in China, great efforts have been made to investigate the epidemic and characterization of B19V in plasma donors, plasma pools, and plasma derivatives in China over the last several years. *Jia et al., (2015)* found that the contamination of B19V in plasma pools was serious in China. In her study, 71.91% (169/235) of plasma pools were contaminated with B19V, with concentrations of 5. $18\times 10^2$–$1.05 \times 10^9$IU/mL. Approximately 31.95% of the DNA-positive plasma pools were only moderately contaminated ($<10^4$IU/mL), while 68.05% contained $>10^4$IU/mL. These data were consistent with those of our study, which demonstrated a relatively high prevalence of B19V in Chinese plasma pools. According to the limit standard of $10^4$ IU/mL established by the US FDA, European Pharmacopoeia, and the Plasma Protein Therapeutics Association (PPTA), and based on our results, approximately 60% plasma pools should be discarded in China, which would be a great waste of plasma. In order to use raw plasma optimally, we recommend B19V DNA screening for individual plasma donors instead of plasma pools and plasma derivatives.

Moreover, we also monitored the contamination of plasma derivatives produced by plasma pools with a viral load of B19V DNA exceeding $10^4$ IU/mL. After considering business privacy, only one company was willing to continue in this study. Therefore, there were only a total of 20 batches of plasma derivatives that were continually collected and monitored. These products were factor VIII, PCC, IVIG, and albumin made from plasma pools with high B19V DNA ($>10^4$IU/mL). The results indicated that B19V DNA was not detected in any batch of albumin and IVIG, except for plasma-derived clotting factor concentrates. The contamination of B19 DNA in plasma-derived clotting factor concentrates was high, indicating that different manufacturing procedures were the key factors. The results were consistent with those of *Geng et al. (2007)*. These contaminated products may impose risks to the patients who received them. There are few clinical data

**Table 3  B19V DNA and antibodies in plasma pools.**

| Blood products manufactures | Sample type | Samples detection (n) | B19V DNA-positive samples | | | | | | | |
|---|---|---|---|---|---|---|---|---|---|---|
| | | | B19 DNA positive (n) | Prevalence of B19 DNA (%) | 95% confidence interval [CI] | B19 viral load $\geq 1 \times 10^4$ IU/mL (n) | Prevalence of B19 viral load $\geq 1 \times 10^4$ IU/mL (%) | 95% confidence interval [CI] | IgG positive (%) | IgM positive (%) |
| A | plasma pools | 100 | 27 | 27 | 18.15 ∼ 5.85 | 4 | 4 | 0 ∼ 6.40 | 100 | 0 |
| B | plasma pools | 100 | 90 | 90 | 84.02 ∼ 95.98 | 21 | 21 | 12.88 ∼ 29.12 | 100 | 0 |
| C | plasma pools | 100 | 81 | 81 | 73.18 ∼ 88.82 | 17 | 17 | 13.74 ∼ 30.26 | 100 | 0 |
| D | plasma pools | 100 | 61 | 61 | 51.27 ∼ 70.73 | 29 | 29 | 19.95 ∼ 38.05 | 100 | 0 |
| Total | plasma pools | 400 | 259 | 64.8 | 60.05 ∼ 69.45 | 71 | 17.8 | 14.91 ∼ 22.59 | 100 | 0 |

**Table 4  Prevalence and levels of B19V DNA and antibodies in plasma derivatives produced by starting plasma pools with viral load $>10^4$ IU/mL.**

| | Start plasma pools | | | Plasma derivatives produced by start plasma pools with viral load $>10^4$ IU/mL | | | | | |
|---|---|---|---|---|---|---|---|---|---|
| Numbers | B19V DNA titers ((IU/mL) | B19V IgG-positive | B19V IgM-positive | Names | Bathes | B19V DNA titers ((IU/mL) | B19V IgG- positive | B19V IgM-positive |
| 1 | $5.25 \times 10^8$ | # | – | | A-1 | N/A | – | – |
| 2 | $9.12 \times 10^7$ | # | – | | A-2 | N/A | – | – |
| 3 | $3.02 \times 10^4$ | # | – | Albumin | A-3 | N/A | – | – |
| 4 | $1.15 \times 10^8$ | # | – | | A-4 | N/A | – | – |
| 5 | $1.60 \times 10^8$ | # | – | | A-5 | N/A | – | – |
| 6 | $9.12 \times 10^7$ | # | – | Intravenous immunoglobulin (pH4) | I-1 | N/A | # | – |
| 7 | $3.02 \times 10^4$ | # | – | | I-2 | N/A | # | – |
| 8 | $1.15 \times 10^8$ | # | – | | I-3 | N/A | # | – |
| 9 | $1.60 \times 10^8$ | # | – | Intravenous immunoglobulin (pH4) | I-4 | N/A | # | – |
| 10 | $3.40 \times 10^7$ | # | – | | I-5 | N/A | # | – |
| 11 | $8.56 \times 10^4$ | # | – | | P-1 | $3.59 \times 10^7$ | – | – |
| 12 | $9.12 \times 10^7$ | # | – | | P-2 | $4.60 \times 10^6$ | – | – |
| 13 | $3.02 \times 10^4$ | # | – | PCC | P-3 | $6.48 \times 10^3$ | – | – |
| 14 | $1.15 \times 10^8$ | # | – | | P-4 | $1.47 \times 10^6$ | – | – |
| 15 | $1.60 \times 10^8$ | # | – | | P-5 | $1.40 \times 10^6$ | – | – |
| 16 | $5.65 \times 10^7$ | # | – | | F-1 | $3.72 \times 10^1$ | – | – |
| 17 | $9.12 \times 10^7$ | # | – | Factor VIII concentrate | F-2 | $1.90 \times 10^1$ | – | – |
| 18 | $1.15 \times 10^8$ | # | – | | F-3 | $1.35 \times 10^2$ | – | – |
| 19 | $1.60 \times 10^8$ | # | – | Factor VIII concentrate | F-4 | $1.65 \times 10^2$ | – | – |
| 20 | $3.40 \times 10^7$ | # | – | | F-5 | $3.57 \times 10^2$ | – | – |

**Notes.**
*B19 IgM positive.
#B19 IgG positive.

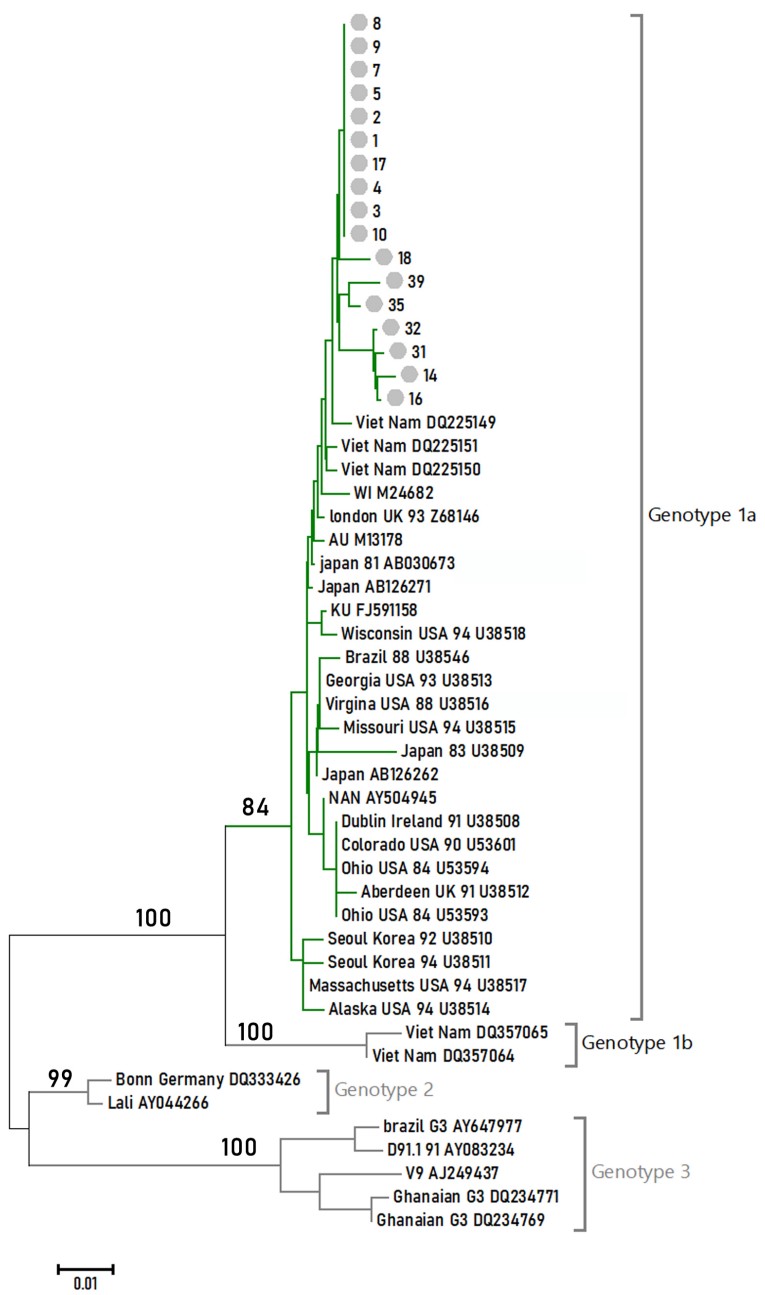

**Figure 1  Phylogenetic relationships of NS1-VP1-U region of different clones.** Values on the nodes indicate NJ/MP bootstrap values. 24 sequences from this study (labeled with Arabic numerals) and a set of reference sequences downloaded from GenBank (labeled with their GenBank accession number) were analyzed.

on the transmission of B19V through plasma derivatives. Previous studies found that the seroprevalence of IgG antibodies to B19V was much higher among very young (age 2–7 years) hemophiliac patients exposed to plasma-derived products compared to those not exposed (*Azzi, Morfini & Mannucci, 1999*; *Santagostino et al., 1997*; *Soucie et al., 2004*).

Another study also found that young children with bleeding disorders exposed only to plasma-derived factor concentrates were 70% more likely to have antibodies to B19V than those who were not exposed to other products (*Soucie et al., 2013*). In that respect, we propose paying more attention to avoid the contamination of plasma-derived clotting factor concentrates.

Previous studies demonstrated that the prevalence of B19V DNA in Chinese plasma donors ranged between 0.03% and 0.09% (*Han et al., 2015*). However, we found that 0.29% (42/14187) of specimens were positive for B19V DNA. This was higher than what was mentioned before, but lower than 0.58%, which *Ke et al. (2011)* found in whole blood donors. The results may be related with the plasma donors' geographic differences and methodological differences in diagnostic procedures. Additionally, only two plasma donors from Guangxi Province were infected with B19V at levels higher than $10^4$IU/mL ($5 \times 10^{12}$ IU/mL). In some reports, the peak virus titer reached $10^{13}$ IU/mL (*Frickhofen & Young, 1989*). Therefore, B19V NAT testing for single Chinese plasma donor screening is necessary. In addition, 11 plasma donors who were B19V DNA-positive at the first screening time were followed up with until the COVID-19 pandemic. One of the plasma donors was a classic example. He was B19V DNA-positive at his first donation, with a virus titer of 5.09 $\times 10^{12}$ IU/mL. After his second and third donation (about 14 days later), he was B19V DNA-positive with a lower virus titer (1.86 $\times 10^5$IU/mL, 1.09 $\times 10^4$IU/mL), and IgG and IgM-positive. This plasma donor may have been in the viremia stage at his first donation. After 36 days, the donors were still B19V DNA and IgG- positive and IgM-negative. Those variations were in accordance with epidemiological trends of B19V in the blood phase. IgM antibodies develop 10–14 days post-infection followed by the development of IgG antibodies directed toward viral capsid components. Meanwhile, this result also indicated that the plasma donor who was infected with B19V could not be permanently refused.

The presence of B19V-specific IgM antibodies suggested recent infection. A survey reported that the seroprevalence of B19V-specific IgM antibodies was commonly below 2% in healthy people. The presence of only B19V-specific IgG antibodies is indicative of past exposure. Both B19V-specific IgG and IgM may be present at or soon after the onset of illness and reach peak titers within 30 days. The prevalence of B19V IgG antibodies increases with age. Other research reported that approximately 2% of children under the age of 5 and 80% of blood donors 18–65 years of age had IgG antibodies (*Ke et al., 2011*; *Kelly et al., 2000*; *Manaresi et al., 2004*). In other studies, about 30% of 18 to 30 year-old donors had detectable IgG, while about 60% of 50 year-old donors were seropositive (*Zaaijer, Koppelman & Farrington, 2004*). A study in our laboratory on the characteristics of Chinese plasma donors showed that most plasma donors (78.5%) were aged 46–55 years old (*Sun et al., 2021*). This may explain the prevalence of B19V-specific IgG in Chinese plasma donors. The Health Council of Netherlands found that blood with persistent anti-B19V IgG (B19V-specific IgG were detected in two separate blood samples, one taken at least 6 months after the other) might be B19V-safe blood (*Groeneveld & van der Noordaa, 2003*). B19V-specific IgM antibodies are detectable 10 to 14 days after infection and can generally persist for 5 months. The prevalence of B19V IgM aided in the assessment of the rate of recent B19V infection in donors. In this study, only one individual B19V DNA-positive

plasma donor was detected with B19V-specific IgM. There was no association between B19V DNA levels and the titer of IgM/IgG yet.

These data recommended the implementation of B19V screening for plasma donors and plasma pools in order to contract the transmission of B19V *via* plasma derivatives in China. Moreover, the National Medical Products Administration prepared the national reference standard for B19V DNA detection and explored the quantitative real-time detection of B19V kits. Whether B19V NAT screening of plasma and plasma derivatives are launched in China or not, Chinese plasma fractionation industries should be encouraged to spontaneously perform B19V NAT screening in plasma donors and mini-pool plasma. These measures can ensure that samples with high titer B19V DNA are discarded in order to control prevention and control of this transfusion-transmitted virus.

B19V is subdivided into three distinct genotypes. *Jia et al. (2016)* found that there were at least three subtypes (1a, 1b, and 3b) of B19V circulating in China. In this study, the prevalent genotypes in 118 B19V-DNA-positive source plasma pool samples attributed 61.86% to genotype 1a, 10.17% to genotype 1b, and 17.80% to genotype 3b. Our study indicated that the prevalent genotypes in the four provinces' plasma donors belonged to B19V genotype 1. These differences may be related to sample resources.

Additionally, our study had some limitations. The plasma derivatives were distributed all around the country and administrated to different recipients, making it difficult to integrate the many resources to perform such large-scale systematic research on the recipients of these blood products. Moreover, participants may have been dropped from the program at any time, and they may have been treated with other blood products. These unfavorable factors may have greatly limited the study. In view of this, B19V NAT screening is recommended in plasma donors and mini-pool plasma to better protect the recipients of plasma derivatives.

## CONCLUSION

In this study, we found that only 0.29% (42/14,187) of plasma donors were positive for B19V DNA, but the contamination of B19V in pooled plasma (64.8%) and plasma-derived clotting factor concentrates (100%) was significant in China. B19V NAT screening is recommended for plasma donors and mini-pool plasma to ensure that samples with high titer B19V DNA are discarded in order to increase the prevention and control of this transfusion-transmitted virus.

## ACKNOWLEDGEMENTS

We are grateful to Professor Miao He for his advice on phylogenetic methods and constructive comments.

### Funding

This study was funded by the Drug Quality Standard Improvement Project from Chinese Pharmacopoeia Commission (2019S01, 2020S04), the Medical Research Program Project

from Sichuan Province (S21072), and the CAMS Innovation Fund for Medical Sciences (2021-1-I2M-060). The funders had no role in study design, data collection and analysis, decision to publish, or preparation of the manuscript.

### Grant Disclosures

The following grant information was disclosed by the authors:
Chinese Pharmacopoeia Commission: 2019S01, 2020S04.
Medical Research Program Project from Sichuan Province: S21072.
CAMS Innovation Fund for Medical Sciences: 2021-1-I2M-060.

### Competing Interests

Xiangzhong Ye, Shangzhi Ji and Jinle Han are employed by Beijing Wantai Biological Pharmacy. Kuilin Lu and Xuexin He are employed by Chengdu Rongsheng Pharmaceutical Co., Ltd. Jiajin Fan and Dawei Cao are employed by Shandong Taibang Biological Products Co., Ltd. Yu Zhang and Yongsheng Yin are employed by Hualan Biological Products Co., Ltd. Yunhua Chen and Xuemei Zhao are employed by Guizhou Taibang Biological Products Co., Ltd.

### Author Contributions

- Pan Sun performed the experiments, authored or reviewed drafts of the article, and approved the final draft.
- Peng Jiang analyzed the data, prepared figures and/or tables, authored or reviewed drafts of the article, and approved the final draft.
- Qing Liu analyzed the data, prepared figures and/or tables, and approved the final draft.
- Rong Zhang analyzed the data, prepared figures and/or tables, and approved the final draft.
- Zongkui Wang conceived and designed the experiments, authored or reviewed drafts of the article, and approved the final draft.
- Haijun Cao conceived and designed the experiments, prepared figures and/or tables, and approved the final draft.
- Xiangzhong Ye performed the experiments, authored or reviewed drafts of the article, and approved the final draft.
- Shangzhi Ji performed the experiments, authored or reviewed drafts of the article, and approved the final draft.
- Jinle Han performed the experiments, prepared figures and/or tables, and approved the final draft.
- Kuilin Lu performed the experiments, authored or reviewed drafts of the article, and approved the final draft.
- Xuexin He performed the experiments, authored or reviewed drafts of the article, and approved the final draft.
- Jiajin Fan performed the experiments, authored or reviewed drafts of the article, and approved the final draft.
- Dawei Cao performed the experiments, authored or reviewed drafts of the article, and approved the final draft.

- Yu Zhang performed the experiments, authored or reviewed drafts of the article, and approved the final draft.
- Yongsheng Yin performed the experiments, authored or reviewed drafts of the article, and approved the final draft.
- Yunhua Chen performed the experiments, authored or reviewed drafts of the article, and approved the final draft.
- Xuemei Zhao performed the experiments, authored or reviewed drafts of the article, and approved the final draft.
- Shengliang Ye analyzed the data, authored or reviewed drafts of the article, and approved the final draft.
- Na Su performed the experiments, authored or reviewed drafts of the article, and approved the final draft.
- Xi Du performed the experiments, analyzed the data, authored or reviewed drafts of the article, and approved the final draft.
- Li Ma conceived and designed the experiments, authored or reviewed drafts of the article, and approved the final draft.
- Changqing Li conceived and designed the experiments, authored or reviewed drafts of the article, and approved the final draft.

## Human Ethics

The following information was supplied relating to ethical approvals (i.e., approving body and any reference numbers):

The Institutional Ethics Review Committee of Institute of Blood Transfusion provided approval to carry out this study (NO: 202029).

## Ethics

The following information was supplied relating to ethical approvals (i.e., approving body and any reference numbers):

The Institutional Ethics Review Committee of Institute of Blood Transfusion approved the study (202029).

## Data Availability

Data is available at NCBI Genbank: OP954001, OP954002, OP954003, OP954004, OP954005, OP954006, OP954007, OP954008, OP954009, OP954010, OP954011, OP954012, OP954013, OP954014, OP954015, OP954016, OP954017.

## Supplemental Information

Supplemental information for this article can be found online at http://dx.doi.org/10.7717/peerj.15698#supplemental-information.

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
