# Peer review of "Parvovirus B19 DNA and antibodies in Chinese plasma donors, plasma pools and plasma derivatives"

_PeerJ, doi:10.7717/peerj.15698_

## Round 0.1 · original submission · Major Revisions

Please address the concerns of all reviewers and amend the manuscript accordingly.

Reviewer 1 ·

Basic reporting

Please refer to comment number 4(additional comments).

Experimental design

Please refer to comment number 4 (additional comments).

Validity of the findings

Please refer to comment number 4 (additional comments).

Additional comments

This study aimed to provide a series of robust evidence on promoting the quality standards for plasma derivatives in China. The samples in the B19V-positive pools and the pooled plasma were tested and then serologic assays and phylogenetic analysis were performed.
I have some comments on this study as follows, with revision, this work will be interesting.
- Keywords arranged alphabetically and selected from the mesh.
- The Objectives of the study are missing in the abstract section. Please write something similar, highlighting the purpose of the study in the abstract section.
- Results section in the abstract should be summarized.
- What reference did you use for phylogenetic analysis?
- The phylogenetic tree should be rooted and used as an outgroup.
- In Material and Methods section-Paragraph “Statistical analysis”- It is not clear on which data the kind of statistical analysis has been applied, please specify and add at the end of material and methods.

·

Basic reporting

The manuscript describes a study on the prevalence of human parvovirus B19 DNA and antibodies in Chinese plasma donors, plasma pools and plasma derivatives. The analyses seem mostly correct and could provide a timely update on the subject for the transfusion medicine community. The manuscript is generally well done, although some sentences would benefit from rephrasing for clarity. Some English language editing is needed.

Experimental design

In the Abstract, “DNA-positive specimens were investigated for the seroprevalence of B19V antibodies”, “individual plasma donors who were B19V DNA positive were followed for a period”, as well as “detection of plasma pools” should be described in “Study design and methods”.

Validity of the findings

L37: Please, change “B19” to “B19V”.
L52: Please clarify “the index time”.
L79-83: The references are needed for “If the prevalence of B19V in blood donors is about 1%, the plasma pools will be contaminated with a high probability. DNA in the blood donor population reaches 1%, their plasma could easily contaminate the entire pooled plasma. Once the single plasma donor with extremely high DNA concentrations exceeding 1014 IU/ml, the B19 viral DNA in the pooled plasma could reach 109 IU/ml.”
L89-90: The sentence “In China, there have been no specific documentation and technical guidelines for monitoring B19V” is repeated with the sentence “However there is no regulation to ensure the safety of pooled plasma and plasma derivatives for B19V in China” in L95-96.
L90: “For better ensuring the safety of plasma donors”? If I have understood correctly, “plasma donors” should change to “plasma derivatives”.
L91: “to perform the screening for B19V in blood and plasma products”? As I know, the current regulatory measures require the detection of B19V DNA in manufacturing pools destined for making some or all kinds of plasma derivatives, other than the detection of plasma products, please confirm.
L115-117: Please specify the follow-up time.

Additional comments

No more.

Reviewer 3 ·

Basic reporting

In general the manuscript is unambiguous, however there are several issues that should be considered before publication.
I suggest revising and proofreading the whole text throughout for proper English language edition. Also, please check that all abbreviations have been explained the first time they are used.
In the abstract, please revise “Study design and methods” and mention serology assays.
The introduction and background show context, nonetheless some references are old and should be replaced with updated cites (examples: (Cotmore and Tattersall 1984; Servant et al 2002). In addition, some passages with important information lack references (for example: Introduction, lines 79-84).
Please revise the methods section as explained below, as well as results and Tables 1, 2 and 3.
The sequences reported in the manuscript should be available/deposited in a public database such as GenBank. Please provide the accession numbers (see figure 1).

Experimental design

-The manuscript peerj-79419 describes an original primary research, referring to a screening study of human parvovirus B19 (B19V) among blood donors of four provinces of China (Shandong, Guangxi, Sichuan and Guizhou), during a ~20-months period (2018-2020). The general aim was to determine the frequency and concentration of B19V DNA in plasma samples and pooled plasma. Authors also intended to provide follow-up data, by evaluating the change in B19V DNA load in plasma of positive donors. In addition, contamination of plasma derivatives as well as B19V-specific immunoglobulins in B19V DNA-positive samples were also analyzed. For that, 17380 individual plasma samples (pools of 48 individuals) from plasmapheresis centers and 400 plasma pools (each pooling approximately 17,000 single plasma samples) from blood products manufacturers were evaluated, assaying B19V DNA by qPCR and sequencing positive samples. To investigate derivatives, batches of albumin, intravenous immunoglobulin, prothrombin-complex concentrate (PCC) and Factor VIII (5 of each) were analyzed when produced with plasma pools with initial B19V DNA >104IU/mL.
-The research question is defined and relevant. The study was performed using standard techniques and ethics in the field. Proceedings are described in general, however some important issues rise and additional specific details should be provided in some cases.
-It is mentioned that B19V DNA-positive samples were confirmed by nested PCR. Why is it necessary to corroborate B19V detection by a nested PCR? Particularly considering that sequence analysis is also performed. Even more: nothing is said in the results section about nested PCR assays in this study.
-Statistical analysis: which tests were performed? What variables are compared and how? Not only this is not specified in the methods section but also no remarks or comments to statistical analysis are made in the results section.
-Individual plasma samples were tested by pools of 48 individuals. What is the expected chance of a 48-individual sample pool results negative missing the opportunity of identifying a low viral load positive individual?
-In the results section, why are blood types described? (lines 161-163).
-It is stated that the presence of anti-B19 antibodies was associated with lower levels of viraemia (line 164). How do authors know this? What is the p-value to support such association? See also lines 283-284 (discussion).
-The follow-up period should be specified. How long and every how many months? (lines 166-167).
-Please consider providing a well elaborated summary of all the important data regarding B19V infection in individuals that were followed-up, not just a "typical example". No details are provided regarding every how many months and how long was each positive individual followed-up, so Table 2 is not appropriate. Authors should include all donors followed-up in the table or a summary (particularly including how long were donors followed-up, what were the minimum and maximum period during which B19V was detected) in the text. See line 169 and Table 2.
-”10 of 11 (83.3%) donors were positive for B19V DNA for no more than 2 times during the following period. 2 donors were positive for B19V DNA all the following period” (lines 169-171). Please revise and correct as appropriate.
-”company B plasma pools with viral load of B19V DNA exceeding 104IU/mL were also collected and tested”. (lines 186-187). This should be explained in the methods section?
-”Table 4 summarized..” (line 187). Table 4 does not provide data summarized, and it should. Please see and revise Table 4.
-Discussion: “more than 60% plasma pools should be discarded in China.” (according to the
limit standard of 10 4 IU/ml) (lines 243-244). Please revise. How this frequency from Jia et al 2015 compares to data from this study (17.8%)? (see Table 3).
-Lines 279-281: “B19V specific IgM antibodies are detectable 10 to 14 days after infection and can generally persist for 5 months”. Please consider discussing the results of the donor in Table 2 in the context of this.

Validity of the findings

-The discussion should be deeply revised and improved (see, for instance, the last paragraph, before conclusions).
-Also carefully revise the conclusions. Please refer to what is concluded from determining the frequency and concentration of B19V DNA in individual plasma and pooled plasma from blood donors from China, evaluating the change in B19V DNA load in plasma of positive donors, contamination of plasma derivatives, and serostatus of positive donors and plasma pools.
-“Further follow-up study on the recipients of these blood products was difficult to perform for us” should not be mentioned since it is not part of the aims of the study.

Additional comments

--

---

## Round 0.2 · Minor Revisions

Please address the remaining concerns of the reviewers and amend your manuscript accordingly.

Reviewer 1 ·

Basic reporting

no comment

Experimental design

no comment

Validity of the findings

no comment

Additional comments

no comment

·

Basic reporting

It is clear that the major criticisms from the initial review have broadly been addressed but there is one issue that still needs to be addressed.

L80-85: B19V deserves discussion mostly because it can circulate at extraordinarily high titers and the extremely high levels of parvovirus B19 in plasma may present a greater risk in plasma derivatives due to pooling of large numbers of plasma units in the manufacture of these products. There is insufficient evidence for these sentences from “If the prevalence of B19V in blood donors is about 1% (Abdelrahman et al. 2021)…” to “the B19 viral DNA in the pooled plasma could reach 109 IU/ml.” And the statement seems not reasonable.

Experimental design

no comment

Validity of the findings

no comment

Reviewer 3 ·

Basic reporting

The manuscript has been revised based on previous suggestions and comments. I still believe that some corrections are needed to improve the English edition, particularly the new paragraphs, such as those in the Discussion section.

Experimental design

No comment.

Validity of the findings

No comment.

---

## Round 0.3 · accepted · Accept

All remaining concerns of the reviewers were adequately addressed and amended manuscript is acceptable now.